# Rough Terrain Navigation Using Divergence Constrained Model-Based Reinforcement Learning

**Sean J. Wang[1], Samuel Triest[2], Wenshan Wang[2], Sebastian Scherer[2], and Aaron M. Johnson[1]**
[1]Department of Mechanical Engineering, [2] Robotics Institute
Carnegie Mellon University
Pittsburgh, PA, USA

**Abstract:** Autonomous navigation of wheeled robots in rough terrain environments has been a long standing challenge. In these environments, predicting the robot's trajectory is challenging due to the complexity of terrain interactions and the divergent dynamics that cause model uncertainty to compound. This inhibits the robot's long horizon decision making capabilities and often lead to short-sighted navigation strategies. We propose a model-based reinforcement learning algorithm for rough terrain traversal that trains a probabilistic dynamics model to consider the propagating effects of uncertainty. Our method increases prediction accuracy and precision by using a tracking controller and by using constrained optimization to find trajectories with low divergence. Using this method, wheeled robots can find non-myopic control strategies to reach destinations with higher probability of success. We show results on simulated and real world robots navigating through rough terrain environments.

**Keywords:** Rough Terrain Navigation, Model-Based Reinforcement Learning, Model Uncertainty

## 1 Introduction

Autonomous wheeled robots are useful for a variety of applications such as environmental monitoring, package delivery, and warehouse operation. Although these robots are capable of autonomously navigating through flat environments, rough terrain still poses a challenge as it requires non-myopic decision making to avoid getting trapped. This is challenging as the interactions between the robot and the unstructured terrain are complex and difficult to model [1]. Furthermore, the highly divergent dynamics can cause modeling errors to compound and propagate poorly along trajectories.

Methods that rely on accurate handcrafted dynamics models for decision making, e.g. [2, 3, 4], often perform poorly when predictions contradict reality. Some methods aim to combat this issue by formulating control policies that are robust against model uncertainty [5, 6, 7, 8]. Others abstract the robot's dynamics by using traversability maps to provide heuristics for how difficult different regions are to traverse [9, 10, 11, 12, 13, 14]. However, these approaches are limited to coarse approximations of traversability and don't consider dynamic interactions.

Alternatively, reinforcement learning can be used for rough terrain traversal. Some methods use model-free reinforcement learning (MFRL) [15, 16, 17], where control policies are directly optimized. Others use model-based reinforcement learning (MBRL) [18], where predictive dynamics models are learned and then used for decision making. Due to their higher sample efficiency, MBRL methods are generally more practical for real world systems where training data is limited and expensive to collect. However, MBRL methods often have poorer performance as their decision making processes may exploit model inaccuracies. Previous MBRL algorithms have addressed this issue by considering prediction uncertainty during decision making [19, 20], though characterizing prediction uncertainty for rough terrain navigation is challenging since the divergent dynamics cause model uncertainty to propagate poorly over long prediction horizons.

We propose a MBRL algorithm for rough terrain navigation, summarized in Sec. 3 and Fig. 1. Our algorithm's unique method of handling uncertainty allows the robot to more accurately optimize

5th Conference on Robot Learning (CoRL 2021), London, UK.

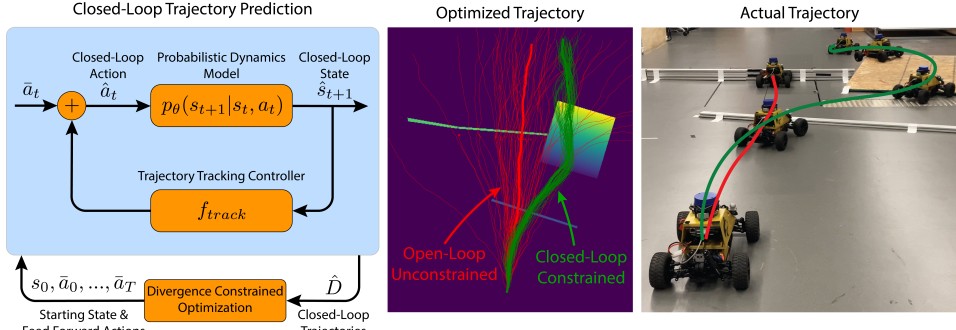

Figure 1: Left: An overview of the method. The probabilistic dynamics model is trained using a multistep loss that considers how uncertainty propagates. The trajectory tracking controller is used to predict a distribution of closed-loop trajectories. The divergence constrained optimizer is used to find a closed-loop trajectory with low divergence. Center: Example trajectory distribution found using our method (green) and a trajectory distribution found without using closed-loop predictions or constrained optimization (red). Right: The actual results of running the two optimized trajectories.

longer trajectories. Like previous methods [19, 20], our method performs trajectory optimization using learned probabilistic dynamics models to predict trajectories and characterize prediction uncertainty. However, we train the probabilistic dynamics model using a multistep loss that considers how model uncertainty propagates along trajectories (Sec. 4). We perform closed-loop trajectory prediction, where the capability of the robot's trajectory tracking controller is considered during trajectory prediction (Sec. 5). The use of a multi-step loss and closed-loop trajectory prediction decreases long horizon trajectory prediction uncertainty and allows the robot to be more selective during optimization. Our algorithm uses constrained optimization to avoid trajectories that result in high divergence (Sec. 6), similar to Convergent Planning [8]. Through simulation and hardware experiments, we show that these improvements allow the robot to find non-myopic trajectories with high prediction accuracy and precision, increasing the likelihood of successful navigation (Sec. 7).

## 2 Related Works

Probabilistic Ensembles with Trajectory Sampling (PETS) [19] aims to make MBRL more robust to modeling errors by characterizing uncertainty with probabilistic ensemble models and propogating uncertainty with trajectory sampling. PETS optimizes trajectories based on expected rewards and performs well on benchmark tasks [21]. However, using PETS on rough terrain is challenging due to the high variance in predictions, which makes it difficult to estimate expected reward. Furthermore, high reward expectation with high variance does not mean the robot will actually perform well.

Kahn et al. [20] use uncertainty in a model-based fashion for collision avoidance in navigation. They train an ensemble of models (via variational dropout [22]) to predict the probability of a collision. They then use this uncertainty to augment their task-specific cost function via a risk-averse prediction of the collision probability, which factors in the uncertainty of the collision predictor ensemble. This collision probability is used in their reward function, which is solved using MPC. This approach is shown to be effective with real-world experiments.

Recent work by Yu et al. [23] takes a more general approach to uncertainty-aware MBRL by learning an ensemble of dynamics models and augmenting arbitrary cost functions with a scaled penalty based on the uncertainty of the ensemble. The authors show that such an approach allows them to perform offline reinforcement learning, or reinforcement learning without access to the environment.

## 3 Method Overview

We formulate the rough terrain navigation problem as a Markov decision process (MDP). Our problem is defined as $(\mathcal{S}, \mathcal{A}, \mathcal{P}, C, \gamma)$, where $\mathcal{S}$ is the state space, $\mathcal{A}$ is the action space, $\mathcal{P}(s'|s, a)$ is the non-deterministic discrete time transition dynamics, $C(s, a)$ is the cost function, and $\gamma$ is the discount factor. We assume that the state includes terrain features (e.g. terrain height map).

Our approach, Fig. 1, falls under the MBRL paradigm. We train a probabilistic neural network to model the unknown transition dynamics $\mathcal{P}$ using previously collected data $\mathcal{D} = \{d_0, ..., d_M\}$, where $d_i = (s_0, .., s_T, a_0, ..., a_{T-1})$ is a state-action trajectory. We train the model to capture both aleatoric and epistemic uncertainty using a multistep training loss that considers how model uncertainty propagates along trajectories (Sec. 4). The trained probabilistic model is used to predict a distribution of the robot's trajectory under the influence of a tracking controller (Sec. 5). Finally, a low cost trajectory to the goal is generated using constrained optimization. The solution is constrained based on the probabilistic model to avoid steps where the dynamics significantly diverges to ensure that the tracking controller can actually achieve the trajectory (Sec. 6).

## 4   Probabilistic Dynamics Models

We propose a method for training a probabilistic dynamics model to predict long horizon trajectories. Similar to [19], we train the model to capture both epistemic and aleatoric uncertainty, however here we propose a multistep loss that considers how prediction uncertainty propagates along a trajectory.

### 4.1   Aleatoric Uncertainty

In the learned model we aim to capture aleatoric uncertainty by modeling the true transition dynamics $\mathcal{P}$ with a probabilistic neural network [24] parameterized by $\theta$. Given a state $s_t$ and action $a_t$ at time $t$, the model predicts a probability distribution of possible next states. For our implementation, the network predicts the mean and log-variance of an uncorrelated gaussian distribution. The estimated likelihood of a next state $s_{t+1}$ conditioned on $s_t$ and $a_t$ is denoted as $p_\theta(s_{t+1}|s_t, a_t)$. The training method in [19] uses a single step likelihood loss. However, this loss fails to consider how prediction uncertainty propagates when multiple single step predictions are chained for trajectory prediction. Furthermore, [25, 26] describe how states generated from earlier predictions may fall out of the distribution of training data, leading to poor predictions for later steps in a trajectory. We adapt [25, 26] to non-deterministic environments by proposing a probabilistic multistep loss that considers the propagation of model uncertainty instead of model error.

The multistep loss we use aims to minimize the negative log likelihood of states conditioned on all prior actions and the starting state of the trajectory, i.e. $\mathcal{L}_{ms}(s_t, A, \theta) := -\log p_\theta(s_t|A)$, where $A = \{s_0, a_0, ..., a_{t-1}\}$. The likelihood of states $s_1, ..., s_t$ conditioned on $A$ can be found as follows:

$$
\begin{aligned}
p_\theta(s_1|A) &= p_\theta(s_1|s_0, a_0) \\
p_\theta(s_2|A) &= \mathbb{E}_{\hat{s}_1 \sim p_\theta(s_1|A)}\big[p_\theta(s_2|\hat{s}_1, a_0)\big] \\
&\;\;\vdots \\
p_\theta(s_t|A) &= \mathbb{E}_{\hat{s}_{t-1} \sim p_\theta(s_{t-1}|A)}\big[p_\theta(s_t|\hat{s}_{t-1}, a_{t-1})\big]
\end{aligned}
\tag{1}
$$

Calculating $\mathcal{L}_{ms}$ exactly is intractable due to the expectations required. Instead we define upper bounds that can be used during training instead, denoted as $\mathcal{L}_{ms}^N$ where $N \in \mathbb{N}$,

$$
\mathcal{L}_{ms}^N(s_t, A, \theta) := -\log \frac{1}{N} \sum_{i=1}^N p_\theta(s_t|\hat{s}_{t-1}^i, a_{t-1})
\tag{2}
$$

where $\hat{s}_j^i$ is sampled recursively from $p_\theta(s_j|\hat{s}_{j-1}^i, a_{j-1})$ for $j < t$. As $N$ approaches infinity, $\mathcal{L}_{ms}^N$ approaches the true multistep loss, $\mathcal{L}_{ms}$. Furthermore, due to Jensen's inequality:

$$
\mathbb{E}\big[\mathcal{L}_{ms}^1(s_t, A, \theta)\big] \geq \mathbb{E}\big[\mathcal{L}_{ms}^2(s_t, A, \theta)\big] \geq ... \geq \mathcal{L}_{ms}(s_t, A, \theta)
\tag{3}
$$

As such, minimizing $\mathcal{L}_{ms}^N$ using any stochastic optimization methods such as Stochastic Gradient Descent (SGD) [27] will effectively also minimize the true loss. Any value could be used for $N$, however larger values give more realistic estimates at the expense of increased computation.

### 4.2   Epistemic Uncertainty

Epistemic uncertainty is defined as model uncertainty due to a poorly distributed or lack of training data. In the context of deep learning, it is typically quantified using a prediction disagreement

metric via an ensemble of neural network models. Some popular methods use explicit ensembles of models [19], while other methods such as Bayesian neural networks [28] or hypernetworks [29, 30] parameterize models via uncertainty in the weights of the neural network.

In our work, we use variational dropout to characterize epistemic uncertainty [31, 22]. With this approach, inputs to the neural network's fully connected layers are randomly dropped (set to zero) with some probability. We define $\Theta$ as the parameters of the neural network dynamics model with no dropout applied. Using dropout on this model will effectively result in a distribution of models. We denote the probability of sampling a model, parameterized by $\theta$, as $q_\Theta(\theta)$.

During training, we randomly sample parameters $\theta$ from $q_\Theta(\theta)$ and calculate the multistep training loss described in Sec. 4.1. We then update $\Theta$ to minimize training loss. The complete model training procedure is summarized in Algorithm 1.

## 5  Closed-Loop Trajectory Prediction

Imprecise trajectory predictions are problematic for two reasons. First, predicted trajectory distributions with high variance provides little insight into the robot's true outcome. Second, the standard error of Monte Carlo predictions scales linearly with prediction variance and inversely with the square root of number of samples. This means that using Trajectory Sampling [19] may require an intractable number of samples when prediction variance is high. We observe that it is common to use a low-level trajectory tracking controllers to correct for deviations of the true trajectory from the desired (predicted) trajectory. Our method aims to predict robot trajectories under the influence of a low-level trajectory tracker controllers. Doing so allows prediction of less divergent robot trajectory distributions and better captures the capabilities of the closed-loop robot control system.

We assume that the controllers take the form:

$$a_t = \bar{a}_t + f_{track}(s_0, ..., s_t, \bar{d}),$$ (4)

where $s_0, ..., s_t$ are the robot's actual states, and $\bar{d} = (\bar{s}_0, ..., \bar{s}_T, \bar{a}_0, ..., \bar{a}_{T-1})$ is a reference trajectory containing reference states and feed forward actions. $a_t$ is the combination of the planned, feed-forward action $\bar{a}_t$ and the corrective action applied to the robot to track the reference trajectory.

Training a dynamics model to directly predict the robot's closed loop behavior is difficult as the controller's inputs are high dimensional and unbounded. Instead, our method first predicts a nominal reference trajectory given a starting state and feed forward actions,

$$\bar{s}_{t+1} = \mathbb{E}\big[s_{t+1} \sim p_\Theta(s_{t+1}|\bar{s}_t, \bar{a}_t)\big].$$ (5)

Our model outputs the predicted distribution's mean which is the expected value above.

A distribution of closed-loop trajectories is then predicted by propagating a set of particles forward in time using the probabilistic dynamics model from Section 4. At each time step, corrective actions for each particle is calculated using $f_{track}$ based on the predicted nominal reference trajectory and the feed forward actions. We denote the predicted closed-loop trajectory distribution as $\hat{D} = \{\hat{d}^0, ..., \hat{d}^I\}$, where $\hat{d}^i = (\hat{s}_0^i, ..., \hat{s}_T^i, \hat{a}_0^i, ..., \hat{a}_{T-1}^i)$ is a particle's state-action trajectory. The procedure for predicting closed-loop trajectories is summarized in Algorithm 2.

## 6  Divergence Constrained Optimization

In MBRL, low cost trajectories are found via trajectory optimization through the learned model. However, naively minimizing cost allows the optimizer to exploit modeling errors, resulting in a trajectory with low predicted cost, but high cost under the true dynamics. Previous work [23] prevented the optimizer from exploiting modeling errors by penalizing trajectories with high prediction uncertainty. This approach may be problematic as the penalty opposes the cost objective. For example, if a robot remains stationary, its trajectory has low prediction uncertainty, but high task cost.

Instead of penalizing uncertainty, our method uses constrained optimization to find low cost trajectories with divergence bounded by some $U$, as trajectory trackers are usually effective as long as the true trajectory does not diverge too much from the reference. Values for $U$ are dependant on the trajectory tracker used. In our experiments, we selected $U$ based on the tracker's observed capabilities.

**Algorithm 1:** Probabilistic Model Training

**for** *number of epochs* **do**
    $\mathcal{L} \leftarrow 0$
    **for** *training batch size(b)* **do**
        $\theta \sim p_\Theta(\theta)$
        sample $A = \{s_0, a_0, ..., a_{T-1}\}$
        $\hat{s}_0^1, ...\hat{s}_0^N \leftarrow s_0$
        **for** $t \in [1, T]$ **do**
            **for** $i \in [1, N]$ **do**
                $\hat{s}_t^i \sim p_\theta(s_t | \hat{s}_{t-1}^i, a_{t-1})$
            $p \leftarrow \frac{1}{N} \sum_{i=1}^{N} p_\theta(s_t^i | \hat{s}_{t-1}^i, a_{t-1})$
            $\mathcal{L} \leftarrow \mathcal{L} - \frac{1}{T} \log p$
    $\Theta \leftarrow$ update using gradient $\nabla_\Theta \frac{\mathcal{L}}{b}$

---

**Algorithm 2:** Closed-Loop Prediction

**Input:** Starting State $s_0$,
            Feed-Forward Actions $\bar{a}_0, ..., \bar{a}_{T-1}$
// Predict Nominal Trajectory
$\bar{s}_0 \leftarrow s_0$
**for** $t \in [1, T]$ **do**
    $\bar{s}_t \leftarrow \mathbb{E}[s_t \sim p_\Theta(s_t | \bar{s}_{t-1}, \bar{a}_{t-1})]$
// Closed-Loop Prediction
**for** $i \in [1, I]$ **do**
    $\theta^i \sim p_\Theta(\theta)$
    $\hat{s}_0^i, \hat{a}_0^i \leftarrow s_0, \bar{a}_0$
    **for** $t \in [1, T]$ **do**
        $\hat{s}_t^i \sim p_{\theta^i}(s_t | \hat{s}_{t-1}^i, \hat{a}_{t-1}^i)]$
        $\hat{a}_t^i \leftarrow \bar{a}_t + f_{track}(\hat{s}_0^i, ..., \hat{s}_t^i, \bar{d})$

---

**Algorithm 3:** Divergence Constrained Optimization

**Input:** Starting State $s_0$
$\bar{a}_0, ..., \bar{a}_{T-1} \leftarrow$ random initialization
**for** *number of Augmented Lagrangian iterations* **do**
    **for** *number of gradient descent iterations* **do**
        $\bar{d}, \hat{D} \leftarrow$ predict nominal and closed-loop trajectories using Algorithm 2
        $u \leftarrow \max_t \sum_i \|\hat{s}_t^i - \bar{s}_t\|_2^2$
        $\mathcal{L} \leftarrow c(\bar{d}) + \lambda \max(u - U, 0)$
        $a_0, ..., a_{T-1} \leftarrow$ update using gradient $\nabla_{a_0,...,a_{T-1}} \mathcal{L}$
    Increase $\lambda$

---

Given a nominal trajectory $\bar{d}$ and predicted closed-loop trajectory distribution $\hat{D}$, we define divergence as follows:

$$u(\bar{d}, \hat{D}) = \max_t \frac{1}{I} \sum_{i=1}^{I} \|\hat{s}_t^i - \bar{s}_t\|_2^2. \tag{6}$$

The constrained optimization problem we aim to solve is defined as follows:

$$\min_{\bar{a}_0,...,\bar{a}_{T-1}} C(\bar{d}) \qquad \text{s.t.} \quad u(\bar{d}, \hat{D}) < U \tag{7}$$

We use an Augmented Lagrangian optimizer to solve (7), as shown in Algorithm 3. During optimization, the estimated divergence is thresholded, $max[u - U, 0]$, scaled by a penalty factor $\lambda$, and added to the nominal trajectory cost. $\lambda$ starts small, but is progressively increased.

## 7 Experiments

We compare our navigation method against others in simulation and in the real world. We used Py-Bullet [32] for our simulation environment and a MuSHR robot [33] for our real world experiment. Simulation training data was collected using random actions (Ornstein–Uhlenbeck noise) in random environments (similar to Figure 3). Real world data was collected using human selected actions (with Gaussian noise) in one environment (similar to Figure 4a with different arrangement). Actions corresponded to motor commands (throttle and steering). Training data consisted of terrain height maps and state-action trajectories. Simulation states included pose (in $SE(3)$) and body velocity ($\mathbb{R}^6$ twist vector). Real world states included pose (in $SE(2)$) estimated using motion capture.

We trained the dynamics models to predict state transition probability, given an action and an observation. Simulation observations included robot tilt ($\mathbb{R}^3$ gravity vector) and body velocity ($\mathbb{R}^6$ twist vector). State transitions consisted of change in pose (in $SE(3)$), and new body velocity. Real world observations included previous state transition. State transitions included change in pose (in $SE(2)$).

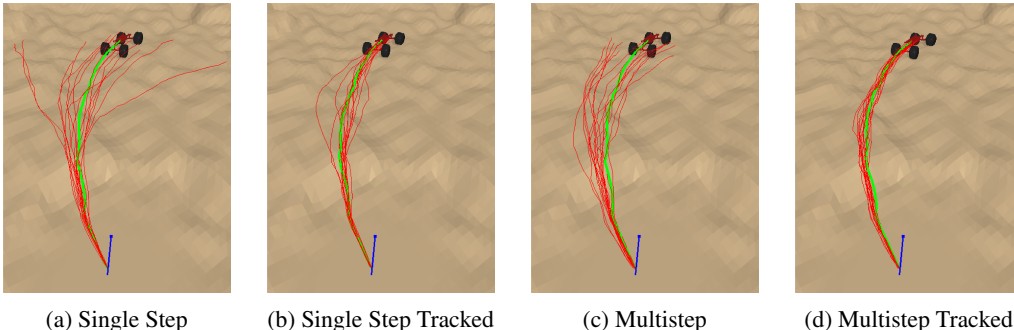

| (a) Single Step | (b) Single Step Tracked | (c) Multistep | (d) Multistep Tracked |

Figure 2: Predicted robot trajectories using different prediction methods The predicted trajectories are shown in red, and the true trajectories are shown in green.

All observations also included a robot centric terrain height map cropped from the world map. Trajectories were predicted by iteratively calculating observations given current states, sampling state transitions from model predictions, and transitioning current states to next states. The models were trained using our proposed multistep loss, $\mathcal{L}_{ms}^{128}$, with Algorithm 1 using 5 step prediction horizons.

We compared three different trajectory optimization frameworks: 1) an optimizer that only minimizes the goal cost, $C(\bar{d})$, 2) an optimizer that adds a scaled divergence penalty, $C(\bar{d}) + \lambda u(\bar{d}, \hat{D})$, and 3) the proposed divergence constrained optimizer (Algorithm 3). We used squared distance to the goal as optimization and evaluation cost. The number of optimization gradient steps were kept consistent between methods. We tracked optimized trajectories using a simple PD controller (detailed definition of $f_{track}$ provided in Sec. B). The effects of the tracker were considering during optimization using closed-loop predictions (Algorithm 2). We treated the threshold $U$ for constrained optimization as a hyperparameter, which we adjusted based on the observed abilities of the tracker.

**Trajectory Prediction Accuracy**

In simulation, we evaluated the effects of using a multistep loss and making closed-loop predictions on trajectory prediction accuracy. We trained one model using a single-step log-likelihood loss, and another model using our proposed multistep loss, $\mathcal{L}_{ms}^{128}$, with Algorithm 1. We used these two models to estimate the probability distribution of the robot's state at the end of a 32 step trajectory by propagating 512 particles using the trained models and averaging the predicted distribution from all particles. We simulated the robot's true trajectory and recorded the log-likelihood of the true state given the estimated distributions. We repeated this process with the closed-loop trajectory predictions using Algorithm 2. The average log-likelihoods over 100 trials are listed in Table 1, which shows an improvement from both the multistep loss and the tracking controller. The true trajectories and some example particle trajectories are shown in Fig. 2.

**Navigating Over Simulated Terrain**

We compared the performance of different navigation methods for the task of driving over randomly generated terrain with large obstacles to a goal $7m$ away. Besides the three aforementioned optimization frameworks, we also included an A* path planning method with tracking using the terrain cost from [34], which considered local terrain smoothness, slope, and curvature. We also evaluated the sensitivity of the optimizers to trajectory prediction accuracy by not performing closed-loop predictions. For 250 trials, Table 2 shows each method's success rate, defined as ending within $0.5m$ of the goal. Table 3 shows the actual cost and standard deviation for each method with tracking.

Overall, we found that the constrained optimization has the highest success rate and lowest final cost. Furthermore, seeding the constrained optimization with the planner resulted in an even higher

| | Single-Step | Single-Step Tracked | Multistep | Multistep Tracked |
|---|---|---|---|---|
| Log-Likelihood | $-692.7 \pm 470.0$ | $-242.5 \pm 364.6$ | $-110.0 \pm 166.6$ | $\mathbf{-31.1 \pm 48.5}$ |

Table 1: Mean and standard deviation of prediction log-likelihood for 100 trials.

|  | A* Planner | No Uncertainty | Penalty | Constraint |
|---|---|---|---|---|
| Tracking | 72 % | 66 % | 60 % | **82 %** |
| No Tracking | - | 62 % | 32 % | 64 % |

Table 2: Random terrain results. Success percentage over fifty random terrains, using the same terrain set for all methods. The A* planner does not provide actions, so tracking was required.

|  | A* Planner | No Uncertainty | Penalty | Constraint |
|---|---|---|---|---|
| Cost (Squared distance to goal) | 2.0892 | 1.8792 | 0.9561 | **0.9109** |
| Divergence | 0.2039 | 0.2350 | **0.1220** | 0.1529 |

Table 3: Results of random terrain testing. Despite having higher divergence than the penalty method, the constrained method has lower final cost.

success rate (88%), suggesting that future work should address issues with local minima. As expected, the penalty method has the lowest divergence since it directly minimizes it, however this did not translate to higher success rate. We also found that prediction accuracy was especially important to uncertainty aware optimization methods. Not including tracking during prediction significantly decreased the success rate of both the penalty and constrained optimization methods.

Figure 3 shows one example trial. When divergence is not considered, the optimizer exploits prediction error and finds a risky solution with high divergence. When executed, the robot collides with an obstacle and rolls over. Conversely, the penalty method finds an overly conservative trajectory since the optimizer minimizes divergence at the expense of task cost. The trajectory has low divergence, but does not reach the goal. The divergence constrained method strikes a middle ground finding a safe low-divergence trajectory to the goal. This behavior was exhibited throughout most of our experiments. Sec E shows more trials and examples.

**Real World Navigation**

We also compared the three optimization frameworks in the real world, Fig. 4. The robot was tasked with navigating across an obstacle field to a goal $5m$ away. The field consisted of a traversable short obstacle, a non-traversable tall obstacle, and a ramp whose traversability depended on the robot's maneuver. Table 4 shows the actual cost and success rate for 10 trials.

When no uncertainty was considered, the optimizer exploited nominal prediction errors and found impossible trajectories through the tall obstacle. For 9 out of 10 trials, the robot got stuck on the tall obstacle. For 1 trial, the true trajectory deviated from the nominal enough for the robot to catch the edge of the ramp and make it to the goal. When penalizing divergence, the optimizer avoided the ramp 6 times to minimize divergence. When using divergence constrained optimization, the robot made it over the ramp 8 times, and on average got closer to the goal.

## 8 Discussion & Conclusion

We propose a novel MBRL algorithm for rough terrain traversal. The main advantage of our method is its ability to accurately optimize long horizon trajectories and thus perform non-myopic decision making. Our method uses divergence constrained optimization to find trajectories with low divergence, ensuring that the robot can faithfully execute the trajectory. However, the benefits of using divergence constrained optimization are less apparent when trajectory predictions are inaccurate. Therefore, our method trains dynamics models to consider uncertainty propagation and uses those models to predict closed-loop trajectories.

|  | No Uncertainty | Penalty | Constraint |
|---|---|---|---|
| Actual Trajectory Cost (Squared distance to goal) | 3.57±1.30 | 5.11±4.45 | **1.68±1.42** |
| Success Rate | 10% | 40% | **70%** |

Table 4: Real world navigation results. Actual cost values are mean and standard deviation for 10 trials. Success is defined as getting within 1.5m of the goal.

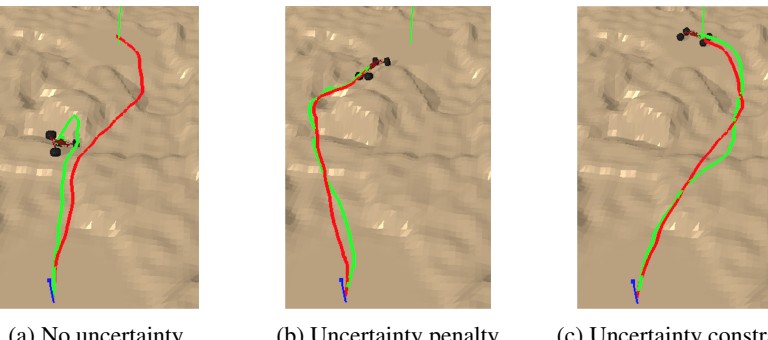

|     |     |     |
| --- | --- | --- |
| (a) No uncertainty | (b) Uncertainty penalty | (c) Uncertainty constraint |

Figure 3: One trial in random terrain. Predicted nominal trajectories are shown in red, and the true trajectories are shown in green. The vertical green lines mark the goal.

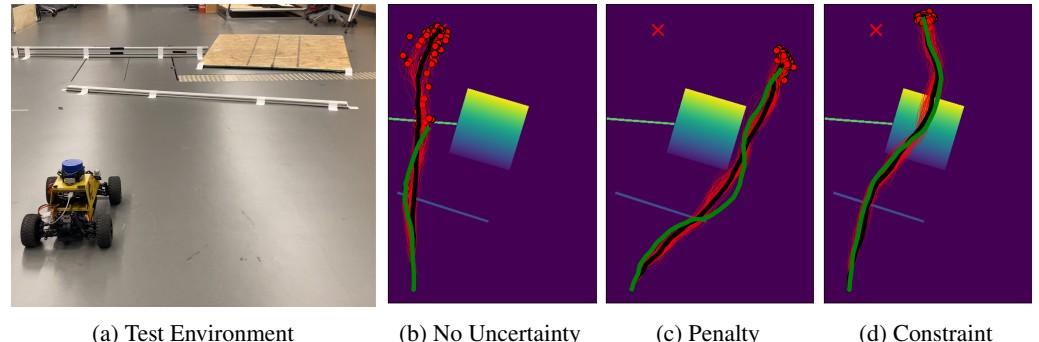

|     |     |     |     |
| --- | --- | --- | --- |
| (a) Test Environment | (b) No Uncertainty | (c) Penalty | (d) Constraint |

Figure 4: One trial of navigating a real world robot across an obstacle field. The goal (red cross), predicted closed-loop trajectories (red lines with last time step marked with dots), predicted nominal trajectory (black line), and actual robot trajectory (green line) are shown over a height map.

Additionally, our method is practical to implement and scalable. First, our method lends itself to offline settings where the agent is unable interact with the environment online. The divergence constrained optimization naturally avoids actions that result in high epistemic uncertainty. As a result, the robot's policy only performs maneuvers that are well supported by the dataset. Second, our algorithm is computationally feasible. Although multiple trajectory predictions are required to propagate uncertainty, this process is easily parallelized for GPU computation. Furthermore, our approach can allow for a much longer horizon when using model predictive control and thus more time for optimization. In our experiments, the algorithm could accurate optimize trajectories with horizons long enough (roughly 100 steps) to not necessitate model predictive control. Lastly, our method is highly modular. Since the dynamics model is agnostic to the tracker used and task's rewards, these components can be changed online without the need to retrain the model.

In the future, improvements to the constrained optimization could increase performance further. In our experiments, the constrained optimization's solution reached a local minimum and that seeding the optimization with trajectories from a planner resulted in higher success. Besides seeding, the optimization could potentially be improved with the use of motion primitives.

In the experiments, robot observations included robot-centric terrain height maps cropped from world height maps. We believe the assumption of world height map knowledge is reasonable since it is common to either build maps a priori or online from sensor data. However, these maps are often imperfect, especially in areas with sparse sensor coverage. Future work will incorporate map uncertainty into trajectory predictions.

Finally, we believe that this approach could also be useful for other tasks. Sec. A describes how the algorithm could be extended to any Partially Observable Markov Decision Process. This could be especially promising for tasks with stochastic dynamics that require non-myopic decision making.

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
