# OpenReview forum: "Rough Terrain Navigation Using Divergence Constrained Model-Based Reinforcement Learning"
_robot-learning.org/CoRL/2021/Conference — CoRL2021 Poster_

### Official Review · Reviewer_79eD · 2021-07-20

**Originality:** Very Good
**Technical Quality:** Excellent
**Clarity Of Presentation:** Very Good
**Impact:** 4

**Recommendation:**

Strong Accept: I recommend accepting the paper and will argue for my recommendation even if other reviewers hold a different opinion.

**Summary:**

This paper presents a method to control wheeled robots to navigate across rough terrain. Uncertainty aware models are trained to model the dynamics and a tracking controller is used to improve prediction accuracy. An constrained optimization is the used to find trajectory with low divergence.

**Issues:**

I think the paper is well written and the proposed method is well executed. No major issues need to be addressed.

**Reviewer Expertise:**

Good: General knowledge of the area

**Strengths And Weaknesses:**

Pro:

1. A model learning algorithm that takes uncertainty into account.

2. A divergence constrained optimization algorithm to find suitable trajectory.

3. Demonstration on a real robot.

Con:

Nothing major.

**Summary Of Recommendation:**

This paper presents a method to control a wheeled robot to navigate across rough terrain. New algorithms are proposed to address issues common in model-based RL and the experiments are well executed, including comparison to alternatives and real robot tests.

---

> ### Author Response · Authors · 2021-08-30
> **Response to 79eD**
>
> We appreciate your review.

---

### Official Review · Reviewer_8wj7 · 2021-07-23

**Originality:** Good
**Technical Quality:** Very Good
**Clarity Of Presentation:** Very Good
**Impact:** 3

**Recommendation:**

Weak Accept: I recommend accepting the paper, but will not argue for my recommendation if the majority of other reviewers have a different opinion.

**Summary:**

This paper is investigating the challenge of using model-based reinforcement learning on rough terrain navigation problems with wheeled robots. The challenge here is that in rough terrain environments with wheels there's a number of different changes in contacts and other dynamics that make prediction difficult and the overall uncertainty of being able to predict into the future larger. The method in the paper then proposes a combination of methods that train over sequences of states into the future to help capture the compounding uncertainties better and preferences in the optimization procedure to avoid using predicted states that have high uncertainty. This combination of methods appears to result in more controled and accurate behavior driving a wheeled robot in a simulated environment and in the real world.

**Issues:**

See the above comments.

**Reviewer Expertise:**

Very good: Comprehensive knowledge of the area

**Strengths And Weaknesses:**

Pros
- The method proposed in the paper appears to be well-founded and the analysis in the paper also shows the positive effects of using each of the components in the overall method.
- The changes for the method are also clean and simple and easy for other people to be able to adopt making the reproduction of the work in this paper simple for others to use.

Cons
- However then part of the challenge here is the methods that are used in this paper have somewhat been explored in other prior work. Then the author in this case needs to better highlight the importance of the unique combination of these methods to be able to solve these particular tasks.
- This method is also primarily validated on the novel wheeled terrain navigation tasks that are shown in the paper. Therefore in order to compensate for some of the cons in the paper, it is important to run the proposed method across the environments that are explored in the PETs paper. This will show that the method also outperforms some other model-based methods and that it's not the case that the new method only performs well on the new custom environment that was created for this paper.

**Summary Of Recommendation:**

See the above comments.

---

> ### Author Response · Authors · 2021-08-30
> **Response to Reviewer 8wj7**
>
> Thank you for your time and feedback. Addressing your concerns has helped make the paper stronger.
>
> We agree the paper could be improved by emphasizing the advantage and importance of using this unique combination of methods. The combination of methods used all aim to improve the robot’s ability to accurately optimize longer horizon trajectories. We have clarified this point in the introduction. Each proposed component of the algorithm is important for this common goal. The constrained optimization allows the robot to find trajectories to the goal that can be executed faithfully. However, this improvement is only noticeable when long horizon trajectory predictions are accurate. Both the multistep loss and closed-loop trajectory predictions contribute to more accurate predictions. We have restructured the experiments section to emphasize this point and added discussions to the discussion & conclusion section.
>
> We agree that a comparison against PETS in OpenAI Gym is useful.The proposed algorithm was created to specifically address the task of rough terrain navigation. However, we do believe that small extensions to the algorithm could make it applicable to any MDP or POMDP problem. We have added discussions on this point in the conclusion, as well as details on how the algorithm could be extended in the appendix.

---

> > ### Comment · Reviewer_8wj7 · 2021-08-31
> > **Responce to reviewer**
> >
> > Hello,
> >
> > I appreciate the response from the reviewers. I understand the method better now and how to place the paper. However, the comments on how the method does not directly work with other tasks have actually increased my concerns about the contribution of the paper. It is stated that the method can not be applied to OpenAIGym tasks or other partially observed tasks because the true initial state is required. This sounds like it decreases the applicability of the method. There are other applications of ML to rough terrain tasks using model-based [1,2,3] and model-free methods [4,5,6] on other morphologies. Is the application of an ML method to a wheeled vehicle rather than other morphologies a significant contribution? Can you describe how the method is novel compared to these other methods that may not require as much information about the robot state?
> >
> > -   [1] P. Fankhauser, M. Bloesch, and M. Hutter, "Probabilistic Terrain Mapping for Mobile Robots with Uncertain Localization", in IEEE Robotics and Automation Letters (RA-L), vol. 3, no. 4, pp. 3019–3026, 2018.
> > -   [2] Grandia, Ruben, et al. "Multi-layered safety for legged robots via control barrier functions and model predictive control." arXiv preprint arXiv:2011.00032 (2020).
> > -   [3] Nguyen, Quan, et al. "Dynamic Walking on Randomly-Varying Discrete Terrain with One-step Preview." Robotics: Science and Systems. Vol. 2. No. 3. 2017.
> > -   [4] Peng, Xue Bin, et al. "Deeploco: Dynamic locomotion skills using hierarchical deep reinforcement learning." ACM Transactions on Graphics (TOG) 36.4 (2017): 1-13.
> > -   [5] Xue Bin Peng, Pieter Abbeel, Sergey Levine, and Michiel van de Panne. 2018. DeepMimic: Example-Guided Deep Reinforcement Learning of Physics-Based Character Skills. ACM Transactions on Graphics (Proc. SIGGRAPH 2018 - to appear) 37, 4 (2018).
> > -   [6] Heess, N., TB, D., Sriram, S., Lemmon, J., Merel, J., Wayne, G., ... & Silver, D. (2017). Emergence of locomotion behaviours in rich environments. arXiv preprint arXiv:1707.02286.

---

> > > ### Author Response · Authors · 2021-09-02
> > > **Response to Reviewer 8wj7**
> > >
> > > Thank you for your response. We have looked through your references and will add relevant citations. We note that most prior work falls into one of two categories. The first focuses on local locomotion and finds actions, footholds, etc. within a short horizon to follow a predefined path [2,3,4,5]. This category assumes that immediate decisions have little effect on the future. In other words, regardless of the robot’s maneuver, it can return to its path or continue towards the goal. The other category plans long horizon paths to the goal using full terrain knowledge, but does so by abstracting the robot’s dynamics. [1] is orthogonal to our work as it focuses on building terrain maps, but follow up work, including the traversability mapping method we used as a baseline [7], falls in this category. While some model-free methods learn policies for long horizon navigation through obstacle fields [6], they are impractical for real world systems and have only been evaluated in simulation. Our algorithm finds long horizon trajectories to the goal based on the robot’s true dynamics and capabilities. Our algorithm is also practical to implement on real world systems. Our real world experiment used 37 minutes of training data collected offline with a human operator.
> > >
> > > While we focused our application on wheeled rough terrain navigation, we do think the approach is applicable to many other tasks. Many tasks, including many OpenAIGym tasks, provide full state feedback. Section A in the appendix also describes how the method could be extended to partially observable tasks such as ones that only provide only image inputs. However, we have not implemented these extensions.
> > >
> > > [7] P. Fankhauser, M. Bjelonic, C. D. Bellicoso, T. Miki, and M. Hutter. Robust rough-terrain 340 locomotion with a quadrupedal robot. In IEEE International Conference on Robotics and 341 Automation, pages 5761–5768, 2018.

---

> > > > ### Comment · Reviewer_8wj7 · 2021-09-03
> > > > **Reponse apprecaited.**
> > > >
> > > > Thank you for the response. I think the method and tasks are well placed inside the related work now. I updated my score.

---

### Official Review · Reviewer_HWwA · 2021-07-24

**Originality:** Very Good
**Technical Quality:** Very Good
**Clarity Of Presentation:** Good
**Impact:** 4

**Recommendation:**

Strong Accept: I recommend accepting the paper and will argue for my recommendation even if other reviewers hold a different opinion.

**Summary:**

Motivated by the problem of building autonomous navigation systems capable of allowing wheeled robots to traverse rough terrain, the authors propose a new system based on model-based reinforcement learning. Specifically, the authors propose to learn a probabilistic dynamics model that provides better predictions of future trajectories (given open-loop control sequences) and also a new planning method in order to select good trajectories. The authors provide interesting real-world experiments that highlight the efficacy of their approach.

**Issues:**

(see "weaknesses" section above)

**Reviewer Expertise:**

Very good: Comprehensive knowledge of the area

**Strengths And Weaknesses:**

STRENGTHS

(S1) The authors study a compelling and difficult problem in robotics, and clearly showcase the benefit of using machine learning in order to approach it.

(S2) The proposed approach (the particular loss function and sampling-based approach to evaluating/deploying) seems novel and has the potential to be quite impactful.

(S3) The authors have performed impressive real-world experimentation.

WEAKNESSES

(W1) The experiment details and metrics used need to be re-hashed.

(a) The authors need to provide more details regarding how training data was collected. In the current submission, only the second paragraph of Section 7 seems to discuss this, and the details are sparse. How much training data must be collected? Are "random actions" really all that are required? It seems doubtful that such an approach would generate diverse enough coverage of the state space to be useful.

(b) "Final cost" is a reported metric, but its unclear as to what that really means for the task at hand. The authors should use a more easily-interpreted metric (time to destination?) in order to measure performance, as "cost" is really whatever the authors define it to be.

(W2) It's not clear from the submission as to what role partial observability plays in the proposed technique. For example, does the proposed approach require a full global map in order to be used? What if this information is not available?

MINOR COMMENTS

(MC1) It's unclear from Figure 3 as to what the difference between (c) and (d) is. This should be highlighted better in the caption.

POST-DISCUSSION COMMENTS

The authors have addressed several of my concerns, especially with respect to the experimental details section. The paper addresses an important topic, and the work seems strong experimentally.

**Summary Of Recommendation:**

I've recommended "weak accept" because I feel that the paper considers and important problem and appears to make a reasonable contribution towards solving it. I do think the authors have a few points to address, but overall the paper seems strong.

---

> ### Author Response · Authors · 2021-08-30
> **Response to Reviewer HWwA**
>
> Thank you for your time and feedback. We have improved the paper based on your concerns.
>
> We agree that the experiment section could benefit from more details. We have added details on the cost metric used in our experiment as well as exactly what “random actions” were taken. In the simulation environment, actions were truly random (OU Noise). In the real environment, actions were controls from a human with added gaussian noise. We have added this to the experiment setup. In both cases, this data was sufficient for training. Our method naturally lends itself well to offline settings, as it naturally avoids behaviors that are not supported by the limited dataset (due to increased epistemic uncertainty). We have added discussion on this point in “Discussion & Conclusion”
>
> We also understand your concern about the availability of global terrain maps. We do believe that this assumption is not unrealistic since it is quite common to either build HD maps a priori or build maps online based on raw sensor data. However, we do believe that these future work should address the issue of uncertainty in the available maps (especially in areas with sparse sensor coverage). We have added discussion on this point in “Discussion & Conclusion”. We have also added discussion in the Appendix on how future work may train dynamics models to make predictions solely based on raw sensor data.
>
> Thank you for the suggestion on Figure 3 (now moved to the appendix). We have made the description more clear. Subfigure c and d show the same trial. Subfigure c shows  the results and d shows the optimization process.

---

### Official Review · Reviewer_stKS · 2021-07-25

**Originality:** Good
**Technical Quality:** Good
**Clarity Of Presentation:** Fair
**Impact:** 3

**Recommendation:**

Weak Reject: I recommend rejecting the paper, but will not argue for my recommendation if the majority of other reviewers have a different opinion.

**Summary:**

The paper presents a model-based RL (MBRL) approach that combines a stochastic state transition model and uncertainty-constrained optimization.
The paper claims to have three new ideas. (1) considering the propagation of model uncertainty along the trajectory (2) combining low-level feedback controller in the trajectory prediction (3) Constrained optimization for uncertainty-bound planning.

The proposed method uses a stochastic state transition model p(s_{t+1}|s_{t}, a_{t}) to predict the evolution of the dynamics. The stochastic model is able to quantify the Aleatoric uncertainty, and the Epistemic Uncertainty is measured by using stochastic dropout. In addition to the naive trajectory prediction, they included a low-level tracking controller (denoted as f_{track} in the paper) in the prediction loop to reduce the variance (This is unclear to me if this is a valid approach).
The estimated uncertainties are then used for planning. Using a constrained optimization solver, they generate the action sequence with the thresholded estimated divergence.

The approach is tested on a wheeled robot both  in simulation and in the real-world. The experimental results & ablation results show that the proposed approach improves the success rate and can be deployed on the real robot.

**Issues:**

- Minimizing loss function in Eq. 2 maximizes the log-likelihood of the state transitions per each step given sampled previous states.
Using a stochastic state transition model and penalizing the whole trajectory prediction is a widely used idea and cannot be seen as a novel idea. For one example, this paper:  "Model-Predictive Policy Learning with Uncertainty Regularization for Driving in Dense Traffic"  ([https://arxiv.org/pdf/1901.02705.pdf](https://arxiv.org/pdf/1901.02705.pdf)).
- The comparison between the "Penalty" method and "Constraint" is not valid. The latter is implemented as an augmented Lagrangian method. The problem is that the augmented Lagrangian optimization is basically the same as the Penalty method in principle. So the paper is comparing a penalty method with a fixed scale to a penalty method with increasing scale.
- I could not find any information about f_{track} in the submission. How is f_{track} defined? Is it only conditioned on the previous states and actions? What happens if the tracking performance differs depending on the terrain conditions?
- The authors claim that the Influence of the low-level controller is considered by looping f_{track} in the trajectory prediction. This is unclear to me. Isn't the influence already learned by the state transition model? The proposed Closed-Loop Prediction can be seen as applying the same controller twice per state transition.
- Typo line 155: does not "divergence" too much.

**Reviewer Expertise:**

Very good: Comprehensive knowledge of the area

**Strengths And Weaknesses:**

- The presented approach is similar to most of the uncertainty-aware MBRL methods except for the "Closed-Loop Prediction" component. I'm unsure about the validity of the "Closed Loop Prediction" part as I argued in the Issues.
- Several important technical details are missing. I could not find any information about the function "f_{track}" which is claimed to be one of the key components of this work. I could not find how the threshold U is defined. As the approach is aimed at the robotics application, I would also be interested in the computational efficiency and sample efficiency of the proposed approach.

**Summary Of Recommendation:**

I admire the effort put into this paper with the real-world experiment and implementation of the whole pipeline. However, the paper seems a bit incomplete. It lacks technical details and some of the claims/ideas are not supported well. (In the initial submission)

---

> ### Author Response · Authors · 2021-08-30
> **Response to Reviewer stKS**
>
> We appreciate the time you have taken to provide us with useful feedback. We have improved the paper based on your concerns.
>
> We believe that the paper’s description of the trajectory tracking controller ($f_{track}$) may have been unclear and led to some confusion. Given a reference trajectory and actions (from the optimization), the trajectory tracker provides corrective actions based on deviation of the robot’s true trajectory from reference trajectory during execution. We did not define the trajectory tracker exactly as our method is agnostic to which trajectory tracker is used. However we have added details on the exact PD trajectory tracker used in our experiments. We believe this example controller could help add clarity to the paper. During offline training data collection, the trajectory tracking controller was not used, as there was no reference trajectory. In our real-world experiment, data was collected while a human operator manually controlled the robot (motor commands from a joystick). In our simulation experiment, data was collected while the robot was driving with random motor commands. If data was to be collected online, the record data would include the actions (commands to the motors), and not the commands given to the trajectory tracker. Since the training data only includes robot motor commands, the predictive dynamics model does not natively predict the effects of the trajectory tracker and it is valid to loop the trajectory tracker controller into trajectory predictions. More experiment details are provided for clarification.
>
> We understand your concern over the novelty using a probabilistic multistep training loss. We realize that it is common to use stochastic transition models, and penalize the robot’s policy based on whole trajectory predictions. This is the case in the paper you cited as well as citations [19,20] from our paper. However, the probabilistic multistep training loss aims to penalize the stochastic state transition model itself during model training and not the policy after the model has been trained. The most similar approach includes [25,26] where a multistep loss is proposed for deterministic systems. However, the authors note concerns that restrict the approach to deterministic systems. We extend the approach to stochastic systems by propagating particles to estimate the aggregated posterior distribution of each state along a trajectory. Our experiments show that this approach leads to more accurate trajectory predictions.
>
> We understand your concern over the similarity between Augmented Lagrangian optimization and fixed-scaled penalty optimization. We agree that Augmented Lagrangian optimizers only approximately solve the constraint optimization problem. However, we believe the fundamental differences between the AL method and the fixed-scale penalty method justify the different classification (constraint vs penalty). The penalty method, which does gradient descent on the uncertainty, essentially solves the optimization problem with a soft equality constraint of $divergence = 0$. Conversely, the AL method solves the optimization problem with an increasingly hard inequality constraint of $divergence <= U$. This fundamental difference mirrors the motivation of using constrained optimization over penalizing divergence during optimization (low amounts of divergence are not problematic, and continuing to decrease divergence below a threshold comes at the expense of higher objective cost).
>
> In general, we agree that more technical details should have been provided in some places. We have added details to the experiment section and appendix. We believe that these details will help clarify some of your concerns.

---

> > ### Comment · Reviewer_stKS · 2021-09-03
> > **Thanks**
> >
> > The technical quality has improved after revision. I raised my score.

---

### Meta-Review · Area_Chair_y5Nb · 2021-08-15

**Recommendation:** Accept (Poster)
**Confidence:** 4

**Metareview:**

The paper a model-based RL (MBRL) approach that takes into account the effect of uncertainty during the planning process. All reviewers agree on the importance of the problem and the good quality of the contribution.

- The paper will benefit from clearly highlighting its key contributions. The paper proposes a combination of improvements in MBRL methods; however, some of these variants have been explored in the literature before. What unique advantages does the proposed combination have? Where are these gains coming from? Which part of the combination? More ablation studies to answer these questions might significantly strengthen the paper.

- A clear discussion on the scalability and applicability of the proposed approach, especially during the uncertainty propagation and the planning phases.

- Demonstration of the proposed approach in simulation (and/or in experiments) on diverse systems and environments will help in understanding the advantages, the key limitations, and the applicability of the proposed approach compared to the state-of-the-art MBRL methods.

Besides addressing the major issues mentioned above, the authors should also revise the manuscript according to the other clarifications
requested and the suggestions provided.

===== Post rebuttal =====

The authors have sufficiently addressed reviewers' concerns; specifically, the discussion on the related work was particularly helpful. I recommend an acceptance.

---

> ### Author Response · Authors · 2021-08-30
> **Response to Area Chair**
>
> We would like to thank all reviewers for their time. We have improved the paper based on your feedback.
>
> We agree that the paper could benefit from a clearer explanation on its key contributions. The proposed MBRL improvements in the paper all lead to the unique advantage of higher prediction accuracy for optimized long horizon trajectories. Without these improvements the predicted trajectories used during decision making are often only accurate within a short horizon, requiring the robot to perform short horizon decision making within an MPC framework. This advantage allows the robot to effectively optimize longer horizon trajectories, which is useful in situations where non-myopic decision making is required. We have better emphasized this unique advantage in the introduction. Our experimental results show that each proposed improvement plays a vital role. Firstly, our comparisons show that using divergence constrained optimization performed better than penalizing or not considering divergence during optimization. However, our experiments show that the improvements due to constrained optimization are less apparent when trajectory predictions are less accurate. We combat this issue through the use of a multistep loss and closed-loop trajectory predictions. Our experiments show that both of these components contribute to more accurate trajectory predictions. We have restructured the experiment section and added discussion to “Discussion and Conclusion” to clarify each component’s importance and performance contributions.
>
> We agree with the importance of scalability and applicability and have added discussion on the matter to “Discussion and Conclusion”. Our approach propagates uncertainty during decision making by making multiple probabilistic predictions of the robot’s trajectory in parallel. Although this approach is more computationally intensive, the process is easily parallelizable to leverage GPUs. Furthermore, our method’s ability to accurately optimize longer trajectories allow for longer MPC horizons, and thus more time for optimization online. In our real world experiment, we were able to accurately optimize a horizon of 100 time steps, eliminating the need for MPC. The discussion in the paper also considers other considerations such as the ability to run offline, and the modularity of the method.
>
> The MBRL algorithm proposed in the paper was created to address the specific task of autonomous rough terrain navigation. However, as we now note in “Discussion & Conclusions”, we believe this algorithm could be extended to any MDP/ POMDP problem. We believe it would be especially useful for tasks that require non-myopic decision making. We have also added details to the appendix on how exactly the algorithm can be extended.

---

### Decision · Program_Chairs · 2021-09-13

**Decision:**

Accept (Poster)

**Comment:**

The paper a model-based RL (MBRL) approach that takes into account the effect of uncertainty during the planning process. All reviewers agree on the importance of the problem and the good quality of the contribution.

- The paper will benefit from clearly highlighting its key contributions. The paper proposes a combination of improvements in MBRL methods; however, some of these variants have been explored in the literature before. What unique advantages does the proposed combination have? Where are these gains coming from? Which part of the combination? More ablation studies to answer these questions might significantly strengthen the paper.

- A clear discussion on the scalability and applicability of the proposed approach, especially during the uncertainty propagation and the planning phases.

- Demonstration of the proposed approach in simulation (and/or in experiments) on diverse systems and environments will help in understanding the advantages, the key limitations, and the applicability of the proposed approach compared to the state-of-the-art MBRL methods.

Besides addressing the major issues mentioned above, the authors should also revise the manuscript according to the other clarifications
requested and the suggestions provided.

===== Post rebuttal =====

The authors have sufficiently addressed reviewers' concerns; specifically, the discussion on the related work was particularly helpful. I recommend an acceptance.